# Health Inequities in LGBT People and Nursing Interventions to Reduce Them: A Systematic Review

**DOI:** 10.3390/ijerph182211801

**Published:** 2021-11-10

**Authors:** Jorge Medina-Martínez, Carlos Saus-Ortega, María Montserrat Sánchez-Lorente, Eva María Sosa-Palanca, Pedro García-Martínez, María Isabel Mármol-López

**Affiliations:** 1Nursing School La Fe, Adscript Centre, University of Valencia, 46026 Valencia, Spain; jorgemquart@gmail.com (J.M.-M.); sanchez_mon@gva.es (M.M.S.-L.); sosa_eva@gva.es (E.M.S.-P.); garcia_pedmarb@gva.es (P.G.-M.); marmol_isa@gva.es (M.I.M.-L.); 2Research Group GREIACC, Health Research Institute La Fe, 46026 Valencia, Spain

**Keywords:** LGBT, sexual and gender minorities, nurses, nursing, social determinants of health, minority stress, systematic review

## Abstract

Background: Lesbian, gay, bisexual, and transgender (LGBT) people present poorer mental and physical health results compared to the heterosexual and cisgender population. There are barriers in the healthcare system that increase these health inequities. Objective: To synthesise the available evidence on how nurses can intervene in reducing health inequities in LGBT people, identifying their specific health needs and describing their experiences and perceptions of the barriers they face in the healthcare system. Methods: Systematic review. Between March and April 2021, a bibliographic search was carried out in the Cuiden, LILACS, PubMed, Dialnet, SciELO, Trip Database, and Web of Science databases and metasearch engines. Inclusion criteria: Articles published in the last 5 years that address the specific health needs of LGBT people, their experiences and perceptions, or interventions in this group in which nurses may engage. Results: A total of 16 articles were selected. Health disparities were detected in the LGBT community, which exhibited higher rates of mental health problems, substance abuse, risky sexual behaviours, self-harm, and suicide. These inequalities were related to minority stress, and each of them differently impacted individual populations within the broader LGBT community depending on their sexual orientations and gender identities. The impact of these factors was, in turn, modified by the intersections of race/ethnicity, geographic region, and socioeconomic factors. LGBT people described discriminatory experiences by health professionals, as well as their distrust and fear in this setting. Nurses can carry out interventions such as inclusive education about sex and sexual and gender diversity and bullying and suicide prevention programmes, and can provide gender-affirming and family-centred care. Conclusions: LGBT people experience health inequities and discrimination in the healthcare system. Nurses can implement diverse interventions to reduce these problems and, moreover, these health professionals are obliged to acquire cultural competence regarding LGBT health.

## 1. Introduction

LGBT is an initialism that refers to the collective of lesbian, gay, bisexual, and transgender people. Other initialisms such as LGBTIAQ (lesbian, gay, bisexual, transgender, intersex, asexual, aromantic, and queer people) or LGBT+ are also often used to explicitly include other groups; however, the acronym LGBT will be used in this article because its use is most widespread worldwide. The term lesbian refers to women who only feel sexual-affective attraction to other women, gay refers to men who only feel sexual-affective attraction to other men, and bisexual refers to people who feel sexual-affective attraction to people of the same or different genders to themselves. A transgender person (often abbreviated as trans) is someone whose gender identity differs from the sex they were assigned at birth: trans women were assigned a ‘male’ identity at birth, but their true gender identity is female; trans men were assigned a ‘female’ identity at birth, but their true gender identity is male; non-binary (NB) is a term that describes people whose gender identity does not fit into the male/female gender binary structure [1,2].

Cis-heteronormativity refers to the assumption that everyone is heterosexual (only attracted to the opposite gender) and cisgender (their gender identity matches the sex they were assigned at birth), as well as the belief that being heterosexual and cisgender (cis-heterosexual) is superior to all other sexual orientations and gender identities. These messages are internalised by all people living in a cis-heteronormative society [1,2,3]. According to the World Health Organization (WHO) [4], health inequities are “avoidable inequalities in health between groups of people within countries and between countries”. Thus, social and economic conditions and their effects on people’s lives determine their risk of illness and their ability to look after their health and to prevent or treat illness [4].

In 2005, the WHO created the Commission on Social Determinants of Health (CSDH), which defined the social determinants of health for the first time in its 2008 report [5]. These are the conditions in which people are born, grow, work, live, and age, as well as the wider set of forces and systems shaping the conditions of daily life [6]. The basic components of the conceptual framework of the social determinants of health include the socio-economic and political context (governance, macroeconomic policies, cultural norms, and societal values), structural determinants (social position, education, occupation, income, race/ethnicity, and gender), and intermediary determinants (material circumstances, psychosocial and biological factors, behaviours, social cohesion, and the healthcare system) [4,6,7].

Various social determinants (social class, gender, race/ethnicity, socioeconomic status, disability, age, geographic location, sexual orientation, and gender identity, among others) contribute (either alone or in concert) to the generation of inequalities, discrimination, marginalisation, and social exclusion, all of which have complex effects on people’s health and well-being [7]. Therefore, intersectionality has been described as a useful tool to analyse these factors and their relationship with health inequities. The term intersectionality was coined by the black feminist lawyer Kimberlé Crenshaw [8] and refers to the study of how different social identities intersect within the systems of power, oppression, and domination. Thus, she posits that each individual has not only one identity but rather, embodies multiple interconnected identities originating from different experiences. In the field of health, the concept of intersectionality is also recognised by the WHO and consists of an analysis of health based on the recognition of social determinants by identifying how power relationships interact at different levels and thereby create health inequities at the individual, institutional, and global levels [2,3,7,9].

According to the available literature [2,10,11,12,13], the LGBT population presents poorer results, both in terms of physical and mental health, compared to the cis-heterosexual population. In relation to health issues, the LGBT community has higher rates of mental health (MH) problems such as depression and anxiety; substance abuse (including the use of tobacco, alcohol, and other drugs); and suicide [2,10,11,12]. The prevalence of osteoporosis and colon, liver, breast, ovarian, or cervical cancers is higher in lesbian and bisexual women and a higher proportion of this population are overweight or obese [10,11]. In gay and bisexual men, higher rates of transmission of the human immunodeficiency virus (HIV), viral hepatitis, and other sexually transmitted infections (STIs); anal, prostate, testicle, and colon cancers; and body image and eating disorders have been described [10,11,12]. In trans people, in addition to their specific needs related to the trans-specific body modification process, high rates of self-harm and suicide have also been reported [10,11,12].

In relation to social issues, trans people experience greater discrimination and high rates of interpersonal violence, while at the same time, fewer people in this population have medical insurance [10,11,12]. Specifically, trans men and NB people with the capacity for pregnancy are often excluded from breast cancer screenings or gynaecological/obstetric care, often because medical staff make the wrong assumptions about the person’s biology [13]. Furthermore, the labour exclusion and poverty experienced by many trans women can lead them into prostitution, exposing them to a greater risk of incarceration, violence, STIs, and drug abuse. Among this sub-group, black and Latin American trans women are the most affected by this type of exclusion and are most susceptible to experiencing physical assault, sexual assault, and murder [10]. In addition, lesbian and bisexual women have a higher risk of not having access to cancer screening services [10,11]. Furthermore, the specific problems of bisexual people are not widely understood because many studies have included this population within the ‘homosexual’ category, thereby rendering them invisible. Nonetheless, based on the available data, it appears that both bisexual women and men are at disproportionate risk of intimate partner violence [10].

Of note, the LGBT community’s health and social needs are different at different stages of their lives. During adolescence and youth, there is a greater risk of suffering from bullying, engaging in risky sexual behaviours, death by suicide, and experiencing family rejection or homelessness [10,11]. In adulthood, higher rates of tobacco, alcohol, and drug abuse have been registered in the LGBT population [10,11]. LGBT people are less likely to have children, and so during old age they are more likely to live alone and to face other health barriers resulting from social isolation and the lack of culturally competent health or social services [10,11]. Moreover, many LGBT people describe feeling uncomfortable and unsafe because of discriminatory attitudes and practices in healthcare systems; they more often report culturally inappropriate care and describe avoiding visiting health centres for the fear of receiving poor care [12,14,15]. Thus, the standardisation of cis-heterosexuality, stereotypes, and prejudices towards the LGBT community impose difficulties in individualised care for this population [12,14,15].

These health inequities can be explained by the minority stress model, first developed by the social worker Virginia Brooks [16,17] and thereafter amplified by the psychiatric epidemiologist Ilan Meyer [18]. This model refers to the stress to which individuals from stigmatised social categories are exposed as a result of their minority social position. These stressors are experienced in addition to the general stressors felt by all people, and are defined as unique, chronic, and socially based [16,18]. They can encompass physical and mental health by causing maladaptive coping strategies and increasing health-risk behaviours, such as rumination, substance abuse, or risky sexual behaviours [19].

The perspective of transcultural nursing developed by the nursing theorist Madeleine Leininger [20] is useful for analysing how nurses’ cultures interfere in the care they provide to various groups. From this theoretical framework, care is understood as a cultural practice while transcultural nursing refers to the comparative study of cultures to understand the similarities and differences between them, seeing these differences as a source of wealth [20]. The goal of transcultural nursing is to provide culturally congruent care that fits each person’s lifestyle, values, beliefs, and meaning system [20]. Along with race/ethnicity and religion, gender identity and sexual orientation can also be considered components of an individual’s culture [12,14,15]. Therefore, this theoretical framework serves to help us understand the LGBT community’s experiences in order to try to avoid biased and discriminatory attitudes during care and to attend to the specific health needs of LGBT people [12,14,15].

The LGBT community faces a series of health inequities derived from the discrimination these individuals experience because of their sexual orientation or gender identity. Both health professionals and the healthcare system can participate in this discrimination or, on the contrary, contribute to its reduction by defending human rights. Therefore, the health inequities present in the LGBT community must be considered both at the individual and structural levels, with the aim of contributing to changes favouring their reduction, thereby helping to achieve health equity.

Nurses are often on the front line of healthcare; they may be an individual’s first point of contact or their primary healthcare provider, and so nurses tend to establish close relationships with patients [12]. They usually provide care to diverse populations living within different social contexts, and the impact of health determinants is incorporated into their practice. An essential component of the nursing role is advocacy, especially for underserved or marginalised populations [12]. Thus, nurses can play key roles in the process of diminishing health disparities in LGBT people.

The research question, in population, intervention, control, and outcomes (PICO) format, that emerged from this literature review was: How can nurses intervene in reducing the health inequities present in LGBT people? Here, we applied a systematic review method to address this research question. Therefore, this study aimed to determine which interventions applied by nurses so far (hereinafter referred to as nursing interventions) have been most effective in reducing the health inequities in LGBT people to date. We aimed to identify the inequalities and specific health needs of the LGBT population, and we describe their experiences and perceptions of the barriers they face in the healthcare system.

## 2. Materials and Methods

### 2.1. Study Design

This study was a systematic review, carried out between March and April 2021, that followed the Preferred Reporting Items for Systematic Reviews and Meta-Analyses (PRISMA) statement [21]. A systematic review is a type of evidence synthesis that uses repeatable methods to collect and analyse secondary data and then identify appropriate data based on a systematic review question.

### 2.2. Search Strategy

Two independent researchers consulted the following electronic databases and metasearch engines: Cuiden, LILACS, PubMed, Dialnet, SciELO, Trip Database, and the Web of Science. For the search strategy, the keywords “LGBT”, “nursing”, and “nurses” were used. These keywords were translated into descriptors using the Descriptores en Ciencias de la Salud (DeCS) [22] and Medical Subject Headings (MeSH) [23] thesauri. The descriptor for “LGBT” was “sexual and gender minorities” (in Spanish, “minorías sexuales y de género”), which were used with the “nursing” and “nurses” descriptors (in Spanish, “enfermería” and “enfermeras”). These terms were combined with the Boolean operators AND OR (Table 1). This article aimed to include every type of nurse (licensed practical nurses, registered nurses, advanced practice nurses, nurse practitioners, etc.), based on the definition of “nurses” according to the MeSH criteria [23].

### 2.3. Selection Criteria

The inclusion criteria were articles published in the last 5 years that addressed the specific health needs of LGBT people, their experiences and perceptions, or nursing interventions in this population. The exclusion criteria were bibliographic reviews; guidelines; articles not written in English, Spanish, Catalan, or Portuguese; and articles with a low level of methodological quality.

### 2.4. Research Variables

The information obtained was grouped based on three variables: the specific health needs of LGBT people (mental and physical health problems), their experiences and perceptions (opinions, satisfaction, discrimination, and recommendations), and nursing interventions in this population (programmes or activities that can be implemented by nurses). These variables were grouped based on the academic literature (thematic analysis). We used The Health of Lesbian, Gay, Bisexual, and Transgender People: Building a Foundation for Better Understanding report by the Institute of Medicine (IOM), ordered by the National Institutes of Health (NIH), as a framework for this literature review [24].

### 2.5. Methodological Quality and the Level of Evidence

Two independent researchers in our group undertook critical reading and assessed the methodological quality of the selected articles by applying the PRISMA statement [21], Critical Appraisal Skills Program Español (CASPe) [25], or the Strengthening the Reporting of Observational studies in Epidemiology (STROBE) statement [26] criteria depending on the type of study. The cut-off points were scores of 17, 6, and 20 for the PRISMA, CASPe, and STROBE statements, respectively. In turn, the level of evidence and the degree of recommendation of the Scottish Intercollegiate Guidelines Network (SIGN) [27] were assessed for each selected article.

### 2.6. The Study Selection Process

The study selection process was as follows: we read the title and abstract of each article; after eliminating any duplicates and applying the inclusion and exclusion criteria, we read the articles in full and evaluated the quality of each one through the lens of the critical reading guidelines. The results of the literature search process are shown in Figure 1.

## 3. Results

After the search, we obtained a total of 381 records, of which 16 articles were selected: 1 systematic review and meta-analysis [28], 5 systematic reviews [29,30,31,32,33], 1 cross-sectional descriptive study [34], 4 qualitative studies [35,36,37,38], and 5 quasi-experimental pre-test–post-test studies [39,40,41,42,43]. Regarding the variables of the reviews, 6 articles addressed the specific health needs of LGBT people [28,29,31,32,33,34], 3 focused on LGBT people’s experiences and perceptions [35,36,37], and 10 presented nursing interventions in the LGBT community [30,31,32,33,38,39,40,41,42,43].

In terms of national contexts, 5 studies were conducted in the USA [38,39,41,42,43], 2 in Canada [36,40], 1 in Ireland [37], 1 in Turkey [35], and 1 in Thailand [34]. According to the classification by the World Bank [44], 8 articles were from high-income economies (the USA, Canada, and Ireland) and 2 were from upper-middle-income economies (Turkey and Thailand). Of the articles selected, 4 studies centred their investigation on LGBT people in general [34,35,36,37], 3 focused on LGBT youth [29,32,33], 1 specifically analysed trans youth [31], and another examined trans women and trans men [28]; 7 studies assessed the training of health professionals and students [30,38,39,40,41,42,43], some of which focused on nurses [40,43] or nursing students [38,41].

### 3.1. The Specific Health Needs of LGBT People

The results of the selected studies [28,29,31,32,33,34] showed that the LGBT population collectively exhibited increased rates of MH problems, substance abuse, risky sexual behaviours, STIs, self-harm, and suicide. These problems were closely associated with experiences related to anti-LGBT attitudes and minority stress such as bullying, isolation, marginalisation, victimisation, and social, family, or peer rejection. Furthermore, LGBT people from racial/ethnic minorities [31,32,33], as well as those who found themselves in poverty [34] or in prostitution [28,31], were more vulnerable to the health disparities mentioned above.

Both bisexual men and women appeared to have a higher risk of MH problems and suicide compared to homosexual people [32,33], and trans people were especially vulnerable to these risks compared to cisgender people [31,32,33]. Among trans people, trans women, particularly those belonging to racial/ethnic minorities and in prostitution, had an increased risk of MH problems, risky behaviours, STIs, or substance abuse because of their social context [28,31]. On the other hand, trans men were less likely to be exposed to STIs than trans women [28].

These studies indicated the presence of LGBT health disparities, which each had a different impact according to individual sexual orientation and gender identity, as well as other factors such as race/ethnicity and socioeconomic factors. Social determinants of health played an essential role in how these inequalities impacted each individual, and these studies highlighted the role of health professionals in recognising and understanding these elements. Nevertheless, it is important to not stereotype or make assumptions about particular patient sub-groups, or presume that LGBT people must necessarily present these health problems.

### 3.2. The Experiences and Perceptions of LGBT People

The results of the selected qualitative studies [35,36,37] showed that LGBT people had reported a lack of information on LGBT health and had suffered discriminatory experiences from health professionals, including the assumption that they were cis-heterosexual and prejudice and stigmatisation for being LGBT or having STIs. The interviewees in some of these studies [35,36,37] described how, when they came out as being LGBT, health professionals usually had prejudices against them or had stereotyped them, making assumptions about their sexual practices or demonstrating negative reactions. Trans people also expressed how they had suffered negative experiences with health professionals in relation to gender transition or by not respecting their name and pronouns [35,36].

### 3.3. Nursing Interventions

Programmes on inclusive sex education, sexual and gender diversity, and bullying and suicide prevention, as well as safe spaces and community-based social groups are highly recommended to improve the MH of LGBT youth in school settings [32,33]. It is also worth highlighting the need for gender-affirming and family-centred care to provide correct information to families about sexual orientation and gender identity, allow them to share their stories, encourage respect, and educate them about the negative health consequences of parental rejection of LGBT youth [31,33]. Likewise, strategies that focus on the specific healthcare of this population should be implemented, in addition to the provision of extra training for health professionals and students [30,31,33,38].

All of the quasi-experimental pre-test–post-test studies included in this review [39,40,41,42,43] agreed that the skills and knowledge related to LGBT cultural competence among health professionals and students significantly increased after they completed educational interventions; their beliefs and support scores also increased, but not significantly [39,41,43] (Table A1). Diverse tools had been used in the context of these educational interventions, including presentations [39,43], standardised patient experiences [38], debriefing sessions [38], meetings [40], e-learning [39,40,42], observational experiences [42], simulations [41], interactive exercises [39], small-group discussions [39], and short films [39]. These methods, together with other approaches such as train-the-trainer programmes and scripted interview sessions or workshops, can be useful to train health professionals and develop cultural awareness of potential health issues related to LGBT people [30].

Nurses can intervene in reducing health disparities within the LGBT community, first by understanding the specific health risks that this population may face. Second, they can also conduct screenings for MH, substance abuse, and adverse childhood experiences; promote a gender-affirming and culturally sensitive health environment by using appropriate pronouns and gender-affirming language on office forms; demonstrate compassionate listening and not make assumptions about any patient; participate in LGBT education or community resource support; provide LGBT health literacy for health professionals; and include all support staff in education programmes to promote an inclusive environment [31]. Third, with regard to policy considerations, nurses can advocate for LGBT health to be included in the nursing school curriculum; support healthcare provider training and education; encourage public health initiatives to engage LGBT people; advocate for provider and consumer competency on LGBT health; and oversee compliance with non-discriminatory policies [31].

A synthesis of the general characteristics of the different studies selected for inclusion in this review is shown in Table 2.

## 4. Discussion

This review provides an overview of the LGBT community’s specific health needs and their experiences within healthcare systems. In addition, some of the articles included also assessed nursing interventions; many of them were quasi-experimental pre-test–post-test studies about educational interventions carried out on health professionals.

### 4.1. The Specific Health Needs of LGBT People

The evidence we reviewed showed that the LGBT population experiences health inequities [28,29,31,32,33,34]. However, the experiences reported in this group were not homogeneous: they varied when multiple marginalised identities were present. Importantly, the bibliography we assessed could be analysed from an intersectional perspective [8]. This standpoint recognises that, when multiple marginalised identities intersect, they represent intertwined inequities. In addition to sexual orientation and gender identity, the health of the LGBT community was influenced by other factors such as gender, race/ethnicity, and social class. Of note, being a woman, belonging to a racial/ethnic minority, or being in poverty or prostitution was related to poorer results in LGBT people, both in terms of physical as well as mental health.

These studies showed that trans women experienced the most exposure to discrimination and violence and had the worst health outcomes [31,32,33]. This fact can be explained by transmisogyny, a term coined by the trans bisexual biologist and writer Julia Serano [45]. Transmisogyny refers to the oppression experienced by trans women (and transfeminine people), in which misogyny and transphobia are interlaced. Likewise, the minority stress model [16,18] was useful to understand some behaviours that may occur in the LGBT community such as substance abuse, risky sexual behaviours, or self-harm. These acts often constitute maladaptive coping mechanisms related to previous negative experiences such as discrimination, victimisation, rejection, anxiety, depression, and loneliness, among others. It is likely that many LGBT individuals use these behaviours, at a general level, to try to regulate high levels of stress and other MH problems derived from stigma that cause them damage.

### 4.2. The Experiences and Perceptions of LGBT People

The evidence shows that LGBT people experience discrimination by some health professionals due to cis-heteronormativity [35,36,37]. The sub-group that reported the most negative experiences consisted of trans people who, for example, were misgendered or deadnamed (being called by the sex pronouns/adjectives or name assigned to them at birth), and many of them avoided or delayed going to a healthcare provider [35,36]. The recommendations for healthcare providers offered by some of the LGBT interviewees were the application of confidentiality and non-judgment; having specific knowledge about LGBT health and gender inclusiveness in care; and the provision of empathetic and person-centred healthcare attention to create a culturally sensitive environment.

The WHO [2] also recognises that there are barriers to accessing the healthcare system for LGBT people because systemic stigma and anti-LGBT attitudes remain obstacles to accessing health services for this population. Indeed, incidents of violence and torture have been documented in healthcare settings, including denial of medical treatment, verbal abuse, forced procedures such as anal exams (to confirm suspicions of anal sex with other men), or so-called ‘sex normalising surgery’ or ‘reparative therapy’.

### 4.3. Nursing Interventions

Nurses can carry out various interventions to reduce health inequities in LGBT people such as programmes on inclusive sex education, sexual and gender diversity, and bullying and suicide prevention [32,33]. In this sense, the figure of the school nurse acquires great importance because of their potential ability to strongly influence outcomes for LGBT people. Gender-affirming and family-centred care were also highlighted in these studies as positive interventions that nurses could apply. Cultural competency training of health professionals and students was identified as one of the most effective interventions [30,31,33,38]; this was closely related to the theoretical framework of transcultural nursing [20].

Educational interventions have been shown to be useful to improve the skills and knowledge related to LGBT cultural competence in health professionals and students [39,40,41,42,43]. However, their scores for beliefs and support did not significantly increase [39,41,43], which may be due to the fact that prejudices are difficult to change once they are established. Therefore, interventions at a younger age, perhaps in schools, could be a more effective approach. Schools are also one of the environments in which experiences of social discrimination are most common [29,31,33,34], making school-focussed interventions indispensable.

The CDC [46] have also offered some recommendations that schools can implement to promote the health and safety of LGBT youth, such as policies that prohibit anti-LGBT attitudes or the creation of support groups and safe spaces for LGBT students. Another suggestion was that information relevant to LGBT youth, and using inclusive language and appropriate terminology, on the prevention of HIV, other STIs, and unwanted pregnancies should be offered in the school curricula. Thus, nurses should adhere to these guidelines to conduct LGBT-inclusive sex education based on the principle of respect.

The Growing Up LGBT+ report [47] also suggested that pupils whose schools had implemented positive messaging about being LGBT had reduced suicidal thoughts and feelings, as well as other MH problems such as panic attacks among members of the LGBT community. In addition, pupils who had received positive messaging about being LGBT felt safer at school. Therefore, positive messaging about sexual and gender diversity may improve the MH and well-being of LGBT pupils.

According to the WHO [2], legal and political barriers must also be addressed at a structural level. This includes the decriminalisation and legal recognition of same-gender relationships and trans people. As such, a WHO guide [48] put forward a series of recommendations related to gay and bisexual men and trans people that included strategies to increase safer sex by implementing both individual and community interventions. Likewise, the report also listed a series of critical enablers for key HIV populations: reviewing laws, policies and practices; reducing stigma and discrimination; empowering the community; and preventing violence. 

The WHO guide [48] also highlighted the importance of creating organisations of LGBT people to promote interaction with other members of the community and thus, generate a greater understanding of the social and health needs of this group. This also encourages both discussion of the consequences of anti-LGBT attitudes and active participation in the provision of comprehensive training on human sexuality. In relation to HIV, the guide recommended that health professionals should respond to the special needs of gay and bisexual men and trans people in a sensitive and empathic manner. Specifically, it stated that nurses could carry out health-risk screening and testing and HIV testing and counselling; initiate and maintain first-line antiretroviral therapy and follow-ups; teach patients about the importance of treatment adherence; and refer individuals to other health and social services.

Importantly, when providing care to LGBT patients, nurses should keep in mind that people who belong to socially marginalised groups may present with mistrust and fear of the healthcare system [12,49]. Culturally sensitive, holistic care is based on treating the patient as a whole person while conducting comprehensive assessments that are aware of the impact of race/ethnicity, social class, sexual orientation, and gender identity, as well as the individual’s past interactions with the healthcare system [49].

Because nurses interact with all health staff, they may be in a privileged position to oversee patient experiences and to lead efforts to educate staff in the fundamentals of providing respectful care to all patients with cultural humility. On occasion, they can advocate for LGBT patients and console or support those who may have been subjected to discriminatory practices [49]. Additionally, nurses can sometimes intervene by convincing patients who have been discriminated against to schedule another appointment with a different provider, or by directing them to patient relations managers to make complaints.

Indeed, in its position statement Nursing Advocacy for LGBTQ+ Populations, the American Nurses Association (ANA) [50] also recognised the need to address LGBT health inequities by advocating for LGBT-inclusive policies and legislation, offering inclusive forms, condemning all discrimination, providing culturally competent care, supporting strategies to educate nurses, collaborating in LGBT education and research, and carrying out interventions aimed at improving the health and wellness of this population.

In summary, all the aforementioned interventions must be approached in an intersectoral manner and carried out at all levels of healthcare provision with the aim of reducing the health inequities experienced by the LGBT community.

### 4.4. Limitations and Future Research

This research is not without limitations. Most of the articles included were from the USA; thus, there was a lack of studies regarding the specific situation in Europe and Spain. All of the studies selected were from English databases; those from databases in Spanish were not included because they did not exactly answer the research question. In addition, very few studies addressed nursing interventions, and many of those that were selected had considered small sample sizes and had used non-randomised convenience sampling, thereby making it difficult to generalise the results. 

Moreover, quasi-experimental pre-test–post-test studies that used self-assessments utilized metrics that are quite subjective and may have had memory or social-desirability biases. Furthermore, the Gay Affirmative Practice Scale (GAP) used in some cases exclusively focused on gay men and lesbian women; hence, bisexual and trans people were not considered in these studies. Thus, future studies that specifically focus on each group within the LGBT community are required. Future lines of research should carry out longitudinal and experimental studies on the interventions used in LGBT communities in order to determine their impact in practice and to establish cause-effect relationships.

## 5. Conclusions

Nurses can carry out interventions such as programmes on inclusive sex education, sexual and gender diversity, and bullying and suicide prevention, as well as provide gender-affirming and family-centred care. In addition, this work highlighted the need for training students and health professionals in LGBT cultural competence. The inequalities endured by—and specific health needs of—LGBT people involve health disparities, higher rates of MH problems, substance abuse, risky sexual behaviours, increased STIs, and a higher rate of self-harm and suicide. These problems were associated with experiences related to anti-LGBT attitudes and minority stress. In turn, the impact of these factors was modified by intersections with race/ethnicity, geographic region, and socioeconomic factors.

LGBT people experience social barriers related to stigma and cis-heteronormativity when trying to access the healthcare system, with many members of this population reporting negative and discriminatory experiences with health professionals. Thus, nurses must learn to recognise and understand the health disparities faced by the LGBT community. Being aware of these inequalities is essential to offer culturally sensitive and gender-affirming care, without making assumptions about any particular patient. Nurses can also participate in LGBT education and community resource support to include all healthcare staff in promoting a safe environment. They can advocate for LGBT health to be included in the nursing school curriculum, support public LGBT health initiatives, oversee compliance with non-discriminatory policies, and console or support patients who may have been subjected to discriminatory practices by other healthcare providers.

## Figures and Tables

**Figure 1 ijerph-18-11801-f001:**
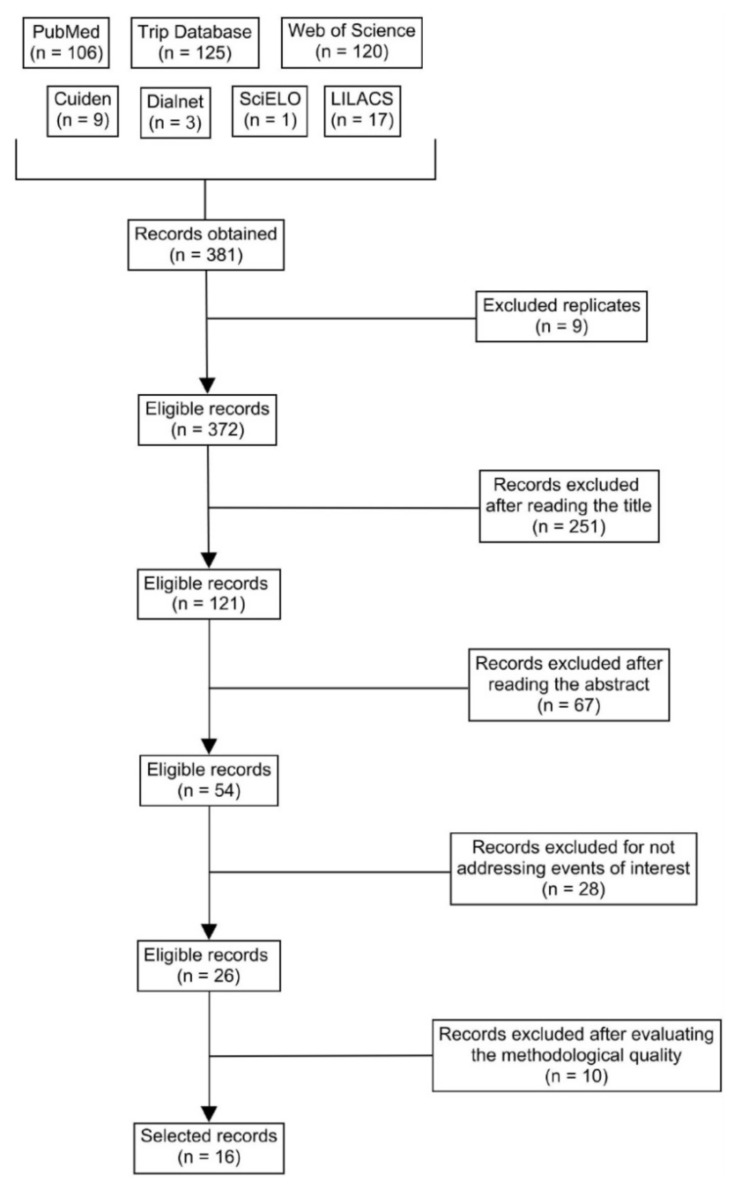
Flow chart of research [21].

**Table 1 ijerph-18-11801-t001:** Database search strategy.

Database/Metasearch Engine	Search String		Obtained Records	Selected Records
Cuiden	Spanish:	((LGBT) OR (minorías sexuales y de género)) AND ((enfermería) OR (enfermeras))	9	0
LILACS			17	0
SciELO			1	0
Dialnet			3	0
PubMed	English:	((LGBT) OR (sexual and gender minorities)) AND ((nursing) OR (nurses]))	106	7
Trip Database			125	3
Web of Science			120	7

**Table 2 ijerph-18-11801-t002:** Synthesis of the results of the studies included in this review.

Authors, Year, and Country	Study Type and Sample	Intervention/Variables	Results/Conclusions	Methodological Quality	Level of Evidence and Grade of Recommendation (SIGN)
Van Gerwen et al. (2020)	Meta-analysis and systematic review. 25 selected articles on trans women and trans men.	The specific health needs of LGBT people.	The prevalence of HIV in trans women was 0–49.6%, and in trans men it was 0–8.3%. The most vulnerable LGBT sub-group was trans women in prostitution. Limitations: the results were difficult to generalise because of wide variability between the studies, and in several cases, only trans women in prostitution were included; there was a lack of studies on trans men and on STIs other than HIV.	PRISMA: 25/27	1+ A
Hossain and Ferreira (2019)	Systematic review. 20 selected articles on LGBT youth.	The specific health needs of LGBT people.	The social context strongly influences the self-conception and self-esteem of LGBT youth. A negative social context favours the appearance of MH problems.	PRISMA: 20/27	2++ B
Raynor et al. (2019)	Systematic review. 10 selected articles on trans youth.	The specific health needs of LGBT people. Nursing interventions.	Factors such as socioeconomic vulnerability, stigma, MH problems, etc., are related to substance abuse and risky behaviours in trans youth. Health professionals must adopt a gender-affirming approach to trans people.	PRISMA: 17/27	2++ B
Wilson and Cariola (2020)	Systematic review. 34 selected articles on LGBT youth.	The specific health needs of LGBT people. Nursing interventions.	Isolation, rejection, anti-LGBT attitudes, and marginalisation, etc., can lead to MH problems such as depression, self-harm, and suicide in LGBT youth. Connectedness to others and implementing specific education programmes in schools can enhance the MH of LGBT youth.	PRISMA: 19/27	2++ B
Yıldız (2018)	Systematic review. 14 selected articles on LGBT youth.	The specific health needs of LGBT people. Nursing interventions.	LGBT youth presented more suicidal ideations and suicide attempts than cis-heterosexual youth. Family-centred care should be one of the principles of practice, and nurses could carry out programmes to prevent suicide and discrimination.	PRISMA: 17/27	2++ B
Kittiteera-sack et al. (2020) Thailand	Descriptive cross-sectional study. 411 LGBT people.	The specific health needs of LGBT people.	Lifetime suicidal ideations were associated with more social discrimination, stress, loneliness, and chronic illnesses. Suicide attempts were associated with internalised anti-LGBT prejudice, poverty, chronic illness, alcohol use, and poor physical health.	STROBE: 20/22	3 D
Logie et al. (2018) Canada	Qualitative study. 51 participants: 16 LGBT teenagers, 21 LGBT adults, and 14 key informants.	The experiences and perceptions of LGBT people.	Cis-heteronormativity and stigma influenced the care provided to LGBT people, who reported negative experiences with health professionals that in turn limited their access to the healthcare system. Non-judgmental care, knowledge of LGBT health, and gender inclusiveness was recommended.	CASPe: 9/10	Q
McCann and Brown (2019) Ireland	Qualitative study. 20 LGBT people.	The experiences and perceptions of LGBT people.	LGBT people reported negative experiences with MH professionals. They expected respectful, empathetic, culturally competent, and person-centred care.	CASPe: 9/10	Q
Karakaya and Kutlu (2020) Turkey	Qualitative study. 18 LGBT people.	The experiences and perceptions of LGBT people.	LGBT people reported having negative experiences with health professionals (stigma for being LGBT or having STIs and assumptions that cis-heterosexuality was the standard, etc.). For fear of discrimination, some LGBT people delayed or avoided accessing the healthcare system.	CASPe: 10/10	Q
McCann and Brown (2018)	Systematic review. 22 selected articles on health professionals.	Nursing interventions.	Training health professionals and inclusion of knowledge about LGBT health in the curricula followed can help these professionals to promote culturally competent care.	PRISMA: 18/27	2++ B
Bristol et al. (2018) USA	Quasi-experimental pre-test–post-test study. 135 health professionals.	Educational intervention. Nursing interventions.	After the intervention, health professionals’ knowledge and skills significantly increased. Openness and support scores also increased, but not significantly. Limitations: use of convenience sampling and low survey return rate.	CASPe: 6/11	2+ C
Du Mont et al. (2020) Canada	Quasi-experimental pre-test–post-test study. 47 forensic nurses.	Educational intervention. Nursing interventions.	After the intervention, both perceived and demonstrated competence significantly increased in all the established domains.	CASPe: 6/11	2+ C
Maruca et al. (2018) USA	Quasi-experimental pre-test–post-test study. 47 nursing students.	Educational intervention. Nursing interventions.	There was a significant difference between the mean pre-test and post-test scores for practical behaviours and a non-significant change in beliefs/attitudes.	CASPe: 6/11	2+ C
Vance et al. (2016) USA	Quasi-experimental pre-test–post-test study. 20 health professionals and students.	Educational intervention. Nursing interventions.	After the intervention, knowledge scores related to trans youth considerably increased.	CASPe: 6/11	2+ C
Wyckoff (2019) USA	Quasi-experimental pre-test–post-test study. 30 nurses.	Educational intervention. Nursing interventions.	There was a statistically significant difference between the total pre-test and post-test scores as well as the behaviour score. In the belief score, the difference was not statistically significant.	CASPe: 6/11	2+ C
Kuzma et al. (2019) USA	Qualitative study. 99 advanced practice nursing students.	Standardised patient experiences and debriefing sessions. Nursing interventions.	The students said that the experiences with standardised patients helped them to develop knowledge and skills on how to treat LGBT patients and set aside their assumptions, prejudices, and biases.	CASPe: 7/10	Q

## Data Availability

Not applicable.

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
