# Peer review of "Health Inequities in LGBT People and Nursing Interventions to Reduce Them: A Systematic Review"

_ijerph, 2021, doi:10.3390/ijerph182211801_

Round 1
Reviewer 1 Report
Overall: This paper addresses an important topic that has been largely neglected in the existing literature. It could be strengthened by addressing concerns identified below and by identifying the unique positions of nurses in healthcare systems and how that may or may not define a unique role for nurses in the provision of, or support of, LGBT-inclusive healthcare.
This systematic review of "nursing interventions" to address LGBT health inequalities, with relatively minor edits, can be a welcome and needed addition to the existing literature on LGBT health disparities and healthcare.
General Comments
1) Considerable evidence suggests that many LGBT health disparities exist, and are perpetuated by, barriers to health care - including avoidance of care due to negative experiences in the receipt of care including encounters with providers who are not trained or willing to provide nondiscriminatory, affirming, respectful, culturally competent and knowledgeable care.
I found this article came up a bit short in specifying specifically what nurses/the nursing profession can uniquely add to the promotion of sexual and gender minority affirming healthcare. The basics of this care begin with visible evidence of SGM inclusiveness in the clinic, such as pamphlets, or nondiscrimination statements and continue with patients' interactions with clinic clerks (how do they call patients to the desk "Mr. Jones? or "Patient Jones"? The first would be offensive if the patient is a transgender woman." LGBT cultural competency continues with initial interviews or filling in forms - are the forms inclusive of LGBT individuals or do LGBT individuals feel that they are not represented among the response options given. Competency continues with the interaction with the first clinical staff person with whom the LGBT person interacts and continues with other health professionals, and hospital staff - including phlebotomy staff and pharmacy staff as well as medical providers. All must be trained to assure culturally competent care.
As nurses are often on the front line of medical care, they have an opportunity to set the tone of the experience - especially if they are also seen again at the end of the appointment visit. Since the basics of LGBT inclusive care are essentially the same for all medical and paramedical personnel, I feel that this article could do a bit more to specify the potential unique role of nurses. As they may interact with clerical, medical, phlebotomy, and pharmacy staff, etc., they may find themselves in a position to oversee/color patients' global experience with their clinic or medical care as well as to organize and lead efforts to educate all relevant staff in the basics of respectful care to all patients, including those who may be sexual and/or gender minorities.
2) A few words about who nurses are and what they do are in order. Does this article aim to include licensed practical nurses, registered nurses, advanced practice nurses, nurse practitioners? I am not sure that there are LGBT-inclusive medical/clinical interventions that are specific to nursing, but it is clear that nurses are well placed to coordinate and promote LGBT inclusive trainings - and this is brought out. I am not sure how this might be added to the paper, but I have witnessed numerous situations in which nurses have consoled patients after having been offended by medical staff (especially in emergency room situations) but also during routine clinic visits. Nurses can sometimes convince offended patients, for example, to schedule another appointment with a different provider or direct them to patient relations. Nurses are also often first-hand witnesses to offensive practices or statements by medical staff and may, at times, support patients who make complaints to patient relations.
3) The article notes Ilan Meyer as the originator of sexual minority stress theory; however, I believe that while writing his thesis, he read the dissertation of Virginia Brooks and amplified a theory that she put forth - and gave her credit. Since in some regions, there is a female predominance in the nursing field, adding a citation to Brooks work would probably come as a welcome surprise to many. The story of her role in minority stress theory can be found here: Rich AJ et al. Sexual minority stress theory: remembering and honoring the work of Virginia Brooks, LGBT Health. 2020 Apr;7(3):124-127. doi: 10.1089/lgbt.2019.0223. Epub 2020 Feb 13.
Specific Comments
1) First para of Abstract and elsewhere: "nursing interventions" may be misconstrued as interventions that are specific to nurses. I recommend, "interventions in which nurses may engage" or something similar. You might spell it out the first time it is used after the abstract followed by "…., here after referred to as nursing interventions."
2) In the abstract and elsewhere, do not refer to high numbers of disorders etc. It would be preferable to speak in terms of "LGBT health disparities with the LGBT population exhibiting higher rates of X, Y and Z, with each of the disparities differentially impacting individual populations within the broader LGBT community depending on their sex assigned at birth and sexual orientation. The impact of these factors is, in turn, modified by their intersections with race, ethnicity, geographic region and socioeconomic factors."
3) Care is needed with use of terminology. In the introduction, for example "Lesbian is a woman who …." This should either be "A lesbian is a…" or Lesbian (italicized) refers to women who…
4) In the first full para of p 2 the meaning of the last sentence is not clear "The LGBT collective includes those people who dissent from heteronormativity in some way."
5) In the 4th paragraph, "Various social determinants …. cause inequalities." The factors do not cause … Consider "these factors, alone, or in interaction contribute to……
6) Bottom of page 2. Consider stating why trans men and NB people "may be excluded from breast and cancer treatments or gynecological/obstetric care." This could be due to discrimination on the basis of sexual or gender minority status, but it is often the result of staff making the wrong assumption about the biology of the person.
7) Rather than saying that the specified factors "lead" transgender women into prostitution, recommend "can lead"
8) Second full paragraph on p 3. Need citation for the first sentence about avoidance of care.
9) Second to last sentence before Materials and Methods - Rather than "what nursing interventions," consider "How can nurses intervene…."
Methods:
1) Due to the time period selected for the literature review, I believe that a seminal work is left out - specifically the 2011 Institute of Medicine Report, The Health of Lesbian, Gay, Bisexual and Transgender People solicited by the NIH in the USA. This work really laid out the approach for LGBT health research that we have been following for the past 10 years and identified most, if not all, relevant factors discussed in this paper.
2) I will leave it to others to critique the literature search. The results are representative of the global literature. Findings have been rather consistent across studies regarding disparities, intersectional factors leading to them, clinical shortcomings and the need for an implementation of LGBT cultural competence training for healthcare professionals and support staff.
Results:
1) Bottom of page 5, section 3.1 and anywhere else the term appears: Please change LGBTphobia to "anti LGBT attitudes." These attitudes are not a phobia/fear.
2) The section on LGBT health disparities is poorly written. It would be better to say that collectively the LGBT population exhibits increased rates of….
3) Bisexual men and women are two separate sexual minority groups and should not be lumped for discussion. Similarly, transgender people should not be lumped. Health and risk profiles are VERY different for transgender men and women.
4) Overall, Section 3.1 is currently inadequately written to convince readers that the authors appreciate the health risks and disparities that differentially impact sexual and gender minority communities. Understanding these is a prerequisite for culturally appropriate work with these populations. Medical staff must understand the unique needs of each different population in a statistical sense, but be prepared to not make assumptions about any particular patient. They must, therefore, understand the range of possibilities but have the cultural sensitivity necessary to ask the medically necessary questions without offending the patient.
5) I am not sure sufficient attention is given to how nurses/others might intervene when patients are refused appropriate care, when they are denied access to hormones or provided with adequate referrals, etc.
6) The role of nurses in education is spelled out - but much of this is in the context of what should be taught. Trainings are limited if trainees obtain high scores but do not practice accordingly. What about nurses' responsibilities in overseeing adherence/compliance with inclusive/nondiscriminatory policies?
7 The findings of the WHO guide are reviewed. What is the role of nurses in implementing these guidelines?
Conclusions
1) This section has a short section with three paragraphs. The first names a few things that nurses can do.
2) The second recapitulates LGBT health inequities
3) The third mentions barriers to care.
4) Will nurses who read this walk away with a strong sense of specifically how they can intervene to address LGBT health disparities? A brief paragraph summarizing such guidance contained within the paper would be helpful.
Author Response
Thank you very much for taking the time to indicate remarks, recommendations and suggestions to improve the article. They were really helpful. We have included them in the manuscript.
General Comments
1) Considerable evidence suggests that many LGBT health disparities exist, and are perpetuated by, barriers to health care - including avoidance of care due to negative experiences in the receipt of care including encounters with providers who are not trained or willing to provide nondiscriminatory, affirming, respectful, culturally competent and knowledgeable care.
I found this article came up a bit short in specifying specifically what nurses/the nursing profession can uniquely add to the promotion of sexual and gender minority affirming healthcare. The basics of this care begin with visible evidence of SGM inclusiveness in the clinic, such as pamphlets, or nondiscrimination statements and continue with patients' interactions with clinic clerks (how do they call patients to the desk "Mr. Jones? or "Patient Jones"? The first would be offensive if the patient is a transgender woman." LGBT cultural competency continues with initial interviews or filling in forms - are the forms inclusive of LGBT individuals or do LGBT individuals feel that they are not represented among the response options given. Competency continues with the interaction with the first clinical staff person with whom the LGBT person interacts and continues with other health professionals, and hospital staff - including phlebotomy staff and pharmacy staff as well as medical providers. All must be trained to assure culturally competent care.
As nurses are often on the front line of medical care, they have an opportunity to set the tone of the experience - especially if they are also seen again at the end of the appointment visit. Since the basics of LGBT inclusive care are essentially the same for all medical and paramedical personnel, I feel that this article could do a bit more to specify the potential unique role of nurses. As they may interact with clerical, medical, phlebotomy, and pharmacy staff, etc., they may find themselves in a position to oversee/color patients' global experience with their clinic or medical care as well as to organize and lead efforts to educate all relevant staff in the basics of respectful care to all patients, including those who may be sexual and/or gender minorities.
Thank you very much for your suggestion. It is true that the article did not focus sufficiently on specific interventions that nurses can carry out. Now, we have tried to explain more how can nurses intervene particularly and we have address the aspects that you have mentioned, which are really important.
For example, it has been included:
“Nurses can intervene in reducing health disparities within the LGBT community, first of all, understanding the specific health risks that these population may face. They can conduct screenings for MH, substance abuse and adverse childhood experiences; promote a gender-affirming and culturally sensitive health environment by using appropriate pronouns and gender-affirming language on office forms, demonstrating compassionate listening and not making assumptions about any patient; participate in LGBT education or community resource support; provide LGBT health literacy for health professionals and include all support staff in education to promote an inclusive environment [30].
With regard to policy considerations, nurses can advocate for LGBT health to be included in the nursing school curriculum, support healthcare provider training and education, support public health initiatives to engage LGBT people, advocate for provider and consumer competency on LGBT health and overseeing compliance with nondiscriminatory policies [30].”
(Results, Section 3.3, page 8, paragraphs 2-3).
Thank you.
2) A few words about who nurses are and what they do are in order. Does this article aim to include licensed practical nurses, registered nurses, advanced practice nurses, nurse practitioners?
Thank you. This article aims to include all type of nurses: licensed practical nurses, registered nurses, advanced practice nurses, nurse practitiones… Culturally competent care and the interventions described should be applied transversely to all nurses.
In this study, we have used the definition of “nurses” according to the Medical Subject Headings (MeSH):
“Professionals qualified by graduation from an accredited school of nursing and by passage of a national licensing examination to practice nursing. They provide services to patients requiring assistance in recovering or maintaining their physical or mental health.”
We have included this in Methods (section 2.2, page 4, paragraph 7):
“This article aimed to include all type of nurses (licensed practical nurses, registered nurses, advanced practice nurses, nurse practitioners…), based on the definition of “nurses” according to the MeSH [22].”
I am not sure that there are LGBT-inclusive medical/clinical interventions that are specific to nursing, but it is clear that nurses are well placed to coordinate and promote LGBT inclusive trainings - and this is brought out. I am not sure how this might be added to the paper, but I have witnessed numerous situations in which nurses have consoled patients after having been offended by medical staff (especially in emergency room situations) but also during routine clinic visits. Nurses can sometimes convince offended patients, for example, to schedule another appointment with a different provider or direct them to patient relations. Nurses are also often first-hand witnesses to offensive practices or statements by medical staff and may, at times, support patients who make complaints to patient relations.
Thank you very much for your suggestion. We have found an article in which explains a case similar to what you have described, where the nurse advocated for the patient, and we have included it (Damaskos, P.; Amaya, B.; Gordon, R.; Walters, C.B. Intersectionality and the LGBT cancer patient. Seminars in Oncology Nursing 2018, 34(1), 30-36, doi:10.1016/j.soncn.2017.11.004).
The explanation says:
“In occasions, they can advocate for LGBT patients, and console or support those who may have received discriminatory practices [48]. Besides, nurses can sometimes intervene by convincing patients who have been discriminated to schedule another appointment with a different provider or by directing them to patient relations to make complaints.”
Thank you.
3) The article notes Ilan Meyer as the originator of sexual minority stress theory; however, I believe that while writing his thesis, he read the dissertation of Virginia Brooks and amplified a theory that she put forth - and gave her credit. Since in some regions, there is a female predominance in the nursing field, adding a citation to Brooks work would probably come as a welcome surprise to many. The story of her role in minority stress theory can be found here: Rich AJ et al. Sexual minority stress theory: remembering and honoring the work of Virginia Brooks, LGBT Health. 2020 Apr;7(3):124-127. doi: 10.1089/lgbt.2019.0223. Epub 2020 Feb 13.
Thank you very much for that information and sorry for the mistake. It really is a welcome surprise that this theory was first developed by a woman, specially in the nursing field. Unfortunately, it seems that her work is not very well-known. It is important to recognize it and give her credit. Now, we have included Virginia Brooks in the article: “These health inequities can be explained by the minority stress model, first developed by the social worker Brooks [16] and afterward amplified by the psychiatric epidemiologist Meyer [17].” (Introduction, page 3. Paragraph 4, lines 1-2).
Specific Comments
1) First para of Abstract and elsewhere: "nursing interventions" may be misconstrued as interventions that are specific to nurses. I recommend, "interventions in which nurses may engage" or something similar. You might spell it out the first time it is used after the abstract followed by "..., here after referred to as nursing interventions."
Thank you for the suggestion. We have changed it. (Abstract, lines 4 and 10; Introduction, p. 4, paragraph 3, lines 2-3).
2) In the abstract and elsewhere, do not refer to high numbers of disorders etc. It would be preferable to speak in terms of "LGBT health disparities with the LGBT population exhibiting higher rates of X, Y and Z, with each of the disparities differentially impacting individual populations within the broader LGBT community depending on their sex assigned at birth and sexual orientation. The impact of these factors is, in turn, modified by their intersections with race, ethnicity, geographic region and socioeconomic factors."
Thank you. It is true. “A high numbers of disorders” is not the best way to express that and it could be misunderstood, it is better to speak in terms of “LGBT health disparities,” which point at social factors. We have changed it following your recommendation. (Abstract, lines 11-16; Results, section 3.1, page 6, paragraph 5, lines 2-3; Conclusions, page 14, paragraph 8).
3) Care is needed with use of terminology. In the introduction, for example "Lesbian is a woman who …" This should either be "A lesbian is a…" or Lesbian (italicized) refers to women who…
Sorry for the mistake. We have corrected it. (Introduction, page 1, lines 5-6 and page 2, line 1).
4) In the first full para of p 2 the meaning of the last sentence is not clear "The LGBT collective includes those people who dissent from heteronormativity in some way."
It is true that that sentece is ambiguous and not very informative, so we have decided to remove it. Thank you for the suggestion.
5) In the 4th paragraph, "Various social determinants … cause inequalities." The factors do not cause … Consider "these factors, alone, or in interaction contribute to…
Thank you. That sentence was not well written. We have replace it with: “Various social determinants (social class, gender, race/ethnicity, socioeconomic status, disability, age, geographic location, sexual orientation, gender identity…), alone or in interaction, contribute to generate inequalities […]” (Introduction, page 2, paragraph 5, lines 1-3).
6) Bottom of page 2. Consider stating why trans men and NB people "may be excluded from breast and cancer treatments or gynecological/obstetric care." This could be due to discrimination on the basis of sexual or gender minority status, but it is often the result of staff making the wrong assumption about the biology of the person.
Thank you for the suggestion. We have added that explanation to specify the cause of why this usually happens to transgender men and NB people: “Specifically, trans men and NB people with capacity for pregnancy are often excluded from breast cancer screenings or gynecological/obstetric care, frequently the result of staff making the wrong assumption about the biology of the person […]” (Introduction, page 3, lines 1-2).
7) Rather than saying that the specified factors "lead" transgender women into prostitution, recommend "can lead".
Thank you. That affirmation (“lead them to prostitution”) may be understood as a generalization that stereotype transgender women. It is important to be careful with the language. It has been replaced with “can lead them to prostitution”, following your recommendation. Sorry for the mistake.
8) Second full paragraph on p 3. Need citation for the first sentence about avoidance of care.
Sorry for the mistake. We have added the citation.
“Also, many LGBT people describe feeling uncomfortable and unsafe due to discriminatory attitudes and practices in healthcare, report culturally inappropriate care or avoid visiting health centers for fear of receiving poor care [12,14,15].” (Introduction, page 3, paragraph 3, lines 1-3).
9) Second to last sentence before Materials and Methods - Rather than "what nursing interventions," consider "How can nurses intervene…."
Thank you for the suggestion. We have change it, as you recommend (Introduction, page 4, paragraph 2, line 2).
Methods:
1) Due to the time period selected for the literature review, I believe that a seminal work is left out - specifically the 2011 Institute of Medicine Report, The Health of Lesbian, Gay, Bisexual and Transgender People solicited by the NIH in the USA. This work really laid out the approach for LGBT health research that we have been following for the past 10 years and identified most, if not all, relevant factors discussed in this paper.
Thank you very much for the suggestion. It is right; due to the time period selected, we did not include the report The Health of Lesbian, Gay, Bisexual and Transgender People, which is fundamental and very informative. Now, we have mentioned it to justify the variables of the study.
“The work The Health of Lesbian, Gay, Bisexual, and Transgender People: Building a Foundation for Better Understanding of the Institute of the Medicine (IOM), requested by the National Institutes of Health (NIH), was used as a framework for this review [23].”
(Methods, section 2.4, page 5, paragraph 1, lines 4-8).
2) I will leave it to others to critique the literature search. The results are representative of the global literature. Findings have been rather consistent across studies regarding disparities, intersectional factors leading to them, clinical shortcomings and the need for an implementation of LGBT cultural competence training for healthcare professionals and support staff.
Thank you very much for those words. We appreciate it. We remain at your disposal.
Results:
1) Bottom of page 5, section 3.1 and anywhere else the term appears: Please change LGBTphobia to "anti LGBT attitudes." These attitudes are not a phobia/fear.
Thank you. It is true, it is more accurate “anti-LGBT attitudes” than “LGBTphobia.” “LGBTphobia” is a term that is quite extended in colloquial speech, but those attitudes do not exist because of phobia/fear but hate. Sorry for the mistake.
2) The section on LGBT health disparities is poorly written. It would be better to say that collectively the LGBT population exhibits increased rates of….
Thank you. We have changed it. Sorry for the mistake.
3) Bisexual men and women are two separate sexual minority groups and should not be lumped for discussion. Similarly, transgender people should not be lumped. Health and risk profiles are VERY different for transgender men and women.
Thank you for the suggestion. Yes, profiles are very different for transgender men and women. We have tried to separate them.
With respect to bisexual people, in the original articles, bisexual people appear lumped, so it is difficult to separate them:
- Yıldız, E. Suicide in sexual minority populations: A systematic review of evidence-based studies. Archives of Psychiatric Nursing 2018, 32(4), 650-659, doi:10.1016/j.apnu.2018.03.003:
“[…] there is an increased risk of suicide attempts in bisexual individuals compared to their homosexual and heterosexual peers.”
“[…] homosexuals and especially bisexual individuals are at risk of suicide […]”
- Wilson, C.; Cariola, L.A. LGBTQI+ youth and mental health: A systematic review of qualitative research. Adolescent Research Review 2020, 5(2), 187-211, doi:10.1007/s40894-019-00118-w.
“In particular, bisexual individuals have been shown to experience more psychological distress, compared to homosexual and heterosexual peers due to experiences of victimization, peer judgments and family rejection.”
4) Overall, Section 3.1 is currently inadequately written to convince readers that the authors appreciate the health risks and disparities that differentially impact sexual and gender minority communities. Understanding these is a prerequisite for culturally appropriate work with these populations. Medical staff must understand the unique needs of each different population in a statistical sense, but be prepared to not make assumptions about any particular patient. They must, therefore, understand the range of possibilities but have the cultural sensitivity necessary to ask the medically necessary questions without offending the patient.
Sorry. We have tried to write the section better, because it wasn’t very well expressed. We have added that health professionals do not have to make assumptions about any patient or stereotype to offer a culturally sensitive care. Thank you for the suggestion.
5) I am not sure sufficient attention is given to how nurses/others might intervene when patients are refused appropriate care, when they are denied access to hormones or provided with adequate referrals, etc.
Thank you for the suggestion. We think that it is a good idea, but, unfortunatelly, we haven’t found interventions in those specific situations. Sorry. However, we have found that nurses can support patients who may have received discriminatory practices, but not those specific contexts: “In occasions, they can advocate for LGBT patients, and console or support those who may have received discriminatory practices [48].”.
6) The role of nurses in education is spelled out - but much of this is in the context of what should be taught. Trainings are limited if trainees obtain high scores but do not practice accordingly. What about nurses' responsibilities in overseeing adherence/compliance with inclusive/nondiscriminatory policies?
Thank you very much for your suggestion. We have added the responsabilities in overseeing compliance with nondiscrimantory policies.
“With regard to policy considerations, nurses can advocate for LGBT health to be included in the nursing school curriculum, support healthcare provider training and education, support public health initiatives to engage LGBT people, advocate for provider and consumer competency on LGBT health and overseeing compliance with nondiscriminatory policies [30].” (Results, Section 3.3, page 8, paragraph 3).
Besides, we have added a position statement of the American Nurses Association (ANA), which also recognizes the role of nurses in advocacy for LGBT people.
“The American Nurses Association (ANA) [49], in its position statement Nursing Advocacy for LGBTQ+ Populations, also recognizes the need to address LGBT health inequities by advocating for LGBT-inclusive policies and legislation, offering inclusive forms, condemning any discrimination, providing a culturally competent care, supporting strategies to educate nurses, collaborating in LGBT education and research, and carrying out interventions aimed at improving the health and wellness of this population.” (Discussion, section 4.3, page 14, paragraph 2).
7) The findings of the WHO guide are reviewed. What is the role of nurses in implementing these guidelines?
Thank you. We have included the role of nurses described in the WHO guide.
“Relating to HIV, this guide recommend that health professionals should respond to the special needs of gay and bisexual men and trans people in a sensitive and empathic manner; specifically, nurses can carry out health risk screening and testing, and HIV testing and counselling, they can also initiate and maintain first-line antiretroviral therapy, follow up, teach about adherence to treatment and refer to other health and social services.” (Discussion, page 13, paragraph 6, lines 9-14).
Conclusions:
4) Will nurses who read this walk away with a strong sense of specifically how they can intervene to address LGBT health disparities? A brief paragraph summarizing such guidance contained within the paper would be helpful.
Thank you for the suggestion. We have added a paragraph to synthesize how nurses can intervene in conclusions.
“It is necessary that nurses recognize and understand the health disparities that face the LGBT community particularly. Being aware of these inequities is essential to offer a culturally sensitive and gender-affirming care, without making assumptions about any particular patient. Nurses can participate in LGBT education or community resource support, including all staff in promoting a safe environment. They can advocate for LGBT health to be included in the nursing school curriculum, support public LGBT health initiatives and overseeing compliance with nondiscriminatory policies, as well as console or support patients who may have received discriminatory practices by other healthcare providers.”
(Conclusions, page 14, paragraph 4).

Reviewer 2 Report
Dear authors,
I appreciated the paper "Health inequities in LGBT people and nursing interventions to reduce them. Systematic review ", which covers an interesting topic.
I provide here some suggestions for improvement. However, I see potential in the study and wish you good luck with your research.
Abstract
The abstract is a bit schematic. I suggest the authors to revise its phrasing to make it more readable.
I would add “systematic review” in the keywords
Introduction
The introduction reports the health and social issues faced by LGBT people. I wonder whether authors could divide those (eg. between health and social issues, ecc.) to ease the reading.
The research question in clear. However, I encourage authors to discuss more in depth in the introduction why the specific nursing field is at the centre of their investigation. Why nurses instead of other categories of health professionals? This requires explanation.
Introduction should mention the method authors used to address the research question.
Method
Section 2.1. is completely underdeveloped. There is plenty of literature explaining the features of systematic review that can be recalled.
381 records were found in databases, while only 16 papers have been selected. The exclusion criteria only refer to bibliographic analyses and other languages, still it seems quite unlikely that these are the only exclusion criteria adopted. What are the exclusion criteria?
As for research variables, authors stated “The information obtained was grouped on three variables: LGBT people’s specific health needs (mental and physical health problems), LGBT people’s experiences and perceptions (opinions, satisfaction, discrimination, recommendations) and nursing interventions (programs or activities that can be carried out by nurses).” How have been those variables selected? It is not clear the methodology followed by authors to group the researches. Is it based on literature (e.g. thematic analysis) or is it derived from a framework (and, if yes, which one?).
Results
Results are presented in synthetic manner. More in depth explanation of the results is required.
Authors could report more information about the results: for instance, in which national contexts were the studies conducted? And, more important, how many paper centred their investigation on lesbians, or gay, or bisexual, trans or nb people? Were there differences in this sense?
The features of table 2 should be described. Table 2 is informative but authors could accompany it by an in depth explanation. Which is the synthesis of the study results (to be reported in the text)?
How are methodological quality and level of evidence assessed?
Concerning educational intervention, which are the most insightful results? Can authors deepen the characteristics of interventions?
Discussion
Discussion starts with the sentence “This review provides an overview of the LGBT population’s specific health needs and their experiences in the healthcare system.” However, the way in which results are presented does not allow the readers to get an overview of the LGBT population’s specific health needs and their experiences in the healthcare system. So, I encourage authors to revise the results section to make it more insightful and understandable and to allow the readers to get a picture of the main evidence in the field.
Conclusion
Conclusions might report better contributions and implications of the research.
Author Response
Thank you very much for taking the time to give us recommendations and suggestions. We have added the changes that you indicated in the manuscript and they have helped improve this article.
Abstract:
The abstract is a bit schematic. I suggest the authors to revise its phrasing to make it more readable.
We have reviewed the abstract profoundly and we have improved its phrasing to make it more readable. Thank you.
I would add “systematic review” in the keywords
Thank you for the suggestion. We have added it.
Introduction:
The introduction reports the health and social issues faced by LGBT people. I wonder whether authors could divide those (eg. between health and social issues, ecc.) to ease the reading.
Thank you for the suggestion. Now, we have divided in health and social issues, because it is more clear.
“[…] In relation to health issues, the LGBT population has higher rates of mental health (MH) problems, such as depression and anxiety; substance abuse, such as tobacco, alcohol and other drugs; as well as suicide [2,10,11,12]. In lesbian and bisexual women, a higher prevalence of osteoporosis, overweight and obesity has been described, and cancers such as colon, liver, breast, ovarian or cervical cancer [10,11]. In gay and bisexual men there are higher rates of transmission of the human immunodeficiency virus (HIV), viral hepatitis and other sexually transmitted infections (STIs), cancers such as anal, prostate, testicle and colon cancer, and disorders of body image and eating disorders [10,11,12]. In trans people, in addition to the needs related to the trans-specific body modification process, there are high rates of self-harm and suicide [10,11,12].
In relation to social issues, trans people experience greater discrimination, high rates of interpersonal violence and lower rates of medical insurance [10,11,12]. Specifically, trans men and NB people with capacity for pregnancy are often excluded from breast cancer screenings or gynecological/obstetric care, frequently the result of staff making the wrong assumption about the biology of the person [13]; on the other hand, the labor exclusion and poverty experienced by many trans women can lead them to prostitution, which exposes them to a greater risk of incarceration, violence, STIs and drug abuse, with black and Latin American trans women being the most affected and the most susceptible to experiencing physical assault, sexual assault and murder [10]. Lesbian and bisexual women have a higher risk of not having access to cancer screening services [10,11]. Furthermore, the specific problems of bisexual people are not widely understood, since in many studies they are included within the category of “homosexual”, being made invisible; however, it appears that both bisexual women and men are at disproportionate risk of intimate partner violence [10].”
(Introduction, page 2, paragraphs 6-7).
The research question in clear. However, I encourage authors to discuss more in depth in the introduction why the specific nursing field is at the centre of their investigation. Why nurses instead of other categories of health professionals? This requires explanation.
We have included the following paragraph that explains why the study is centered in nurses:
“Nurses are often on the front line of healthcare; they may be an individual’s first point of contact or primary provider of healthcare and they establish a close relationship with the patient [12]. Usually, they provide care to diverse populations who live within different social contexts, and the impact of determinants of health are incorporated into their practice. An essential component of the nursing role is advocacy, especially for underserved or marginalized populations [12]. For these reasons, nurses may be a key piece in the process of diminish LGBT health disparities.”
(Introduction, page 3, paragraph 8).
Thank you.
Introduction should mention the method authors used to address the research question.
Thank you. We have added the method in the introduction.
“The research question in PICO format that emerges from the literature review is: How can nurses intervene in reducing health inequities in LGBT people?
The method used to address the research question is a systematic review.”
(Introduction, page 4, paragraphs 2-3).
Method:
Section 2.1. is completely underdeveloped. There is plenty of literature explaining the features of systematic review that can be recalled.
Sorry for the mistake. We have added the characteristics of a systematic review and the PRISMA statement, used to make this article.
“This study is a systematic review carried out between March and April 2021. This review followed the Preferred Reporting Items for Systematic Reviews and Meta-Analyses (PRISMA) statement [20]. A systematic review is a type of evidence synthesis that uses repeatable methods to collect secondary data and analyze it, and then identify data based on a systematic review question.”
(Methods, section 2.1, page 4).
381 records were found in databases, while only 16 papers have been selected. The exclusion criteria only refer to bibliographic analyses and other languages, still it seems quite unlikely that these are the only exclusion criteria adopted. What are the exclusion criteria?
Sorry for the mistake. We have added that we also excluded guidelines and studies with low methodological quality (the cut-off point for exclusion is in the section 2.5).
As for research variables, authors stated “The information obtained was grouped on three variables: LGBT people’s specific health needs (mental and physical health problems), LGBT people’s experiences and perceptions (opinions, satisfaction, discrimination, recommendations) and nursing interventions (programs or activities that can be carried out by nurses).” How have been those variables selected? It is not clear the methodology followed by authors to group the researches. Is it based on literature (e.g. thematic analysis) or is it derived from a framework (and, if yes, which one?).
Sorry. We have added that the variables were selected based on literature and the framework. Thank you for the suggestion.
“These variables were grouped based on literature (thematic analysis). The work The Health of Lesbian, Gay, Bisexual, and Transgender People: Building a Foundation for Better Understanding of the Institute of the Medicine (IOM), requested by the National Institutes of Health (NIH), was used as a framework for this review [23].” (Methods, section 2.4, paragraph 1, lines 4-8).
Results:
Results are presented in synthetic manner. More in depth explanation of the results is required.
Authors could report more information about the results: for instance, in which national contexts were the studies conducted? And, more important, how many paper centred their investigation on lesbians, or gay, or bisexual, trans or nb people? Were there differences in this sense?
The features of table 2 should be described. Table 2 is informative but authors could accompany it by an in depth explanation. Which is the synthesis of the study results (to be reported in the text)?
Thank you for your suggestions. Yes, the section of results was a bit schematic. We have addded the aspects that you noted and we have extended the section of results with more information. All the changes are marked in red.
How are methodological quality and level of evidence assessed?
Sorry. We have included methodological quality and level of evidence in the section 2.5.
“A critical reading was performed and the methodological quality of the articles was assessed by two independent researchers using the PRISMA statement [20], the Critical Appraisal Skill Program Español (CASPe) [24] or the STROBE statement [25] depending on the type of study. The cut-off point for the PRISMA statement was a score of 17; for the CASPe, 6; and for the STROBE statement, 20.
In turn, the level of evidence and the degree of recommendation of the Scottish Intercollegiate Guidelines Network (SIGN) [26] of each selected article were assessed.”
Concerning educational intervention, which are the most insightful results? Can authors deepen the characteristics of interventions?
Thank you. Now, we have written a paragraph explaining the characteristics of interventions.
“[…] Within the characteristics of the educational interventions, diverse tools were used: presentations [38,42], standardized patient experiences [37], debriefing sessions [37], meetings [39], e-learning [38,39,41], observational experiences [41], simulations [40], interactive exercises [38], small-group discussions [38] or short films [38]. These methods, together with other like train-the-trainer programs, scripted interview sessions or workshops, can be useful to train health professionals and develop cultural awareness of potential health issues related to LGBT people [37].”
(Results, section 3.3, page 7).
Discussion:
Discussion starts with the sentence “This review provides an overview of the LGBT population’s specific health needs and their experiences in the healthcare system.” However, the way in which results are presented does not allow the readers to get an overview of the LGBT population’s specific health needs and their experiences in the healthcare system. So, I encourage authors to revise the results section to make it more insightful and understandable and to allow the readers to get a picture of the main evidence in the field.
Sorry. We have changed the section of results and have added more information to allow the readers to visualize the current evidence in the field. We hope that now it is more clear and informative. Thank you.
Conclusion:
Conclusions might report better contributions and implications of the research.
Thank you for your suggestion. We have added a paragraph that sumarizes the implications of the research.
“It is necessary that nurses recognize and understand the health disparities that face the LGBT community particularly. Being aware of these inequities is essential to offer a culturally sensitive and gender-affirming care, without making assumptions about any particular patient. Nurses can participate in LGBT education or community resource support, including all staff in promoting a safe environment. They can advocate for LGBT health to be included in the nursing school curriculum, support public LGBT health initiatives and overseeing compliance with nondiscriminatory policies, as well as console or support patients who may have received discriminatory practices by other healthcare providers.”
(Conclusions, page 14, paragraph 4).

Round 2
Reviewer 2 Report
Dear authors,
the paper has improved since its first version and all the comments have been addressed.
I do not have additional comments.
Wish you the best.
Author Response
Dear reviewer,
Thank you very much for taking the time to revise the manuscript. We are really grateful.
The manuscript has been reviewed by an official translator from our department, as Reviewer 1 suggested. We attach certification.
Yours sincerely.
